# Effects of Dietary Bioactive Lipid Compounds of *Acacia nilotica* Bark on Productive Performance, Antioxidant Status, and Antimicrobial Activities of Growing Rabbits under Hot Climatic Conditions

**DOI:** 10.3390/ani13121933

**Published:** 2023-06-09

**Authors:** Ahmed A. A. Abdel-Wareth, Hazem G. M. El-Sayed, Hamdy A. Hassan, Ghadir A. El-Chaghaby, Abdel-Wahab A. Abdel-Warith, Elsayed M. Younis, Shimaa A. Amer, Sayed Rashad, Jayant Lohakare

**Affiliations:** 1Department of Animal and Poultry Production, Faculty of Agriculture, South Valley University, Qena 83523, Egypt; hamdy_ahmed@agr.svu.edu.eg; 2Regional Center for Food and Feed, Agricultural Research Center, Giza 12619, Egypt; h.elsayed@arc.sci.eg (H.G.M.E.-S.); ghadiraly@yahoo.com (G.A.E.-C.); sayed_rashad79@hotmail.com (S.R.); 3Department of Zoology, College of Science, King Saud University, P.O. Box 2455, Riyadh 11451, Saudi Arabia; awarith@ksu.edu.sa (A.-W.A.A.-W.); emyounis@ksu.edu.sa (E.M.Y.); 4Department of Nutrition and Clinical Nutrition, Faculty of Veterinary Medicine, Zagazig University, Zagazig 44511, Egypt; shimaa.amer@zu.edu.eg; 5Poultry Center, Cooperative Agricultural Research Center, Prairie View A&M University, Prairie View, TX 77446, USA; jalohakare@pvamu.edu

**Keywords:** *Acacia nilotica*, antioxidants, antibacterial, sustainability, rabbits

## Abstract

**Simple Summary:**

Feeding strategies to increase production efficiency using novel feed additives are necessary for sustainable rabbit production. This study aims to use *Acacia nilotica* bark bioactive lipid compounds (ANBBLCs) as novel feed additives in the growing rabbit diet for sustainable production and to improve antioxidant status and antibacterial activity during hot climatic conditions. ANBBLCs up to 150 mg/kg can be used as an efficient new feed additive to increase growth performance, carcass criteria, antioxidant status, as well as the antibacterial activity of growing rabbits under hot climatic conditions.

**Abstract:**

This study aimed to evaluate the efficacy of dietary *Acacia nilotica* bark bioactive lipid compounds (ANBBLCs) as novel feed additives on the growth performance, carcass criteria, antioxidants, and antimicrobial activities of growing male rabbits. A total of 100 California male weanling rabbits aged 35 days were divided into four nutritional treatments, each of which contained ANBBLCs at concentrations of 0 (control group), 50, 100, and 150 mg/kg diet (*n* = 25 per treatment, each replication consisting of one animal). The average body weight of the animals was 613 ± 14 g. The experiments lasted for 56 days. Dietary ANBBLC levels linearly improved (*p* < 0.05) the body weight, body weight gain, and feed conversion ratio (FCR) of rabbits. Furthermore, with increasing concentrations of ANBBLCs, the total antioxidant capacity of blood and liver tissue was linearly (*p* < 0.05) enhanced. *Lactobacillus* increased and *Staphylococcus* decreased (*p* < 0.05) in comparison to the control group when ANBBLC levels were added to the diets of rabbits. Rabbit diets supplemented with ANBBLCs increased dressing percentages and decreased abdominal fat. This study shows that ANBBLCs can be used as a feed additive to enhance the growth performance, carcass criteria, antioxidant, and antibacterial properties of growing rabbits.

## 1. Introduction

The two primary issues facing meat production in Egypt are still food security and hot climatic conditions. The usage of phytogenic chemicals as feed additives is a typical nutritional technique to boost overall revenue, promote food security, and decrease the issues related to hot environmental conditions. Rabbit diets can be supplemented with plant bioactive lipid compounds (BLCs), which are frequently but inaccurately referred to as “essential oils” [1], in an effort to improve the health, growth, and productivity of the animals. Because they have few functional sweat glands and have difficulties sweating out extra body heat, rabbits are highly susceptible to high temperatures [2], which can affect their metabolism, performance, carcass metrics, and meat quality [3]. 

The *Acacia nilotica* tree belongs to the Mimosaceae family and is commonly cultivated in subtropical and tropical Africa [4,5]. It is an important plant in urban, agricultural, and pastoral systems. The majority of this plant’s components have proven to be highly beneficial in traditional therapies. Diverse bioactive secondary ingredients have been found in this plant, which are involved in its diverse biological functions, including gallic acid, isoquercitin, terpenes, phenolic glycosides, volatile essential oils, ascorbic acid, carotene, calcium, magnesium, and selenium [6].

Natural compounds derived from plants provide opportunities for research due to their important pharmacological and toxicological properties, and sometimes they are considered to be novel medications for the treatment of tumors, inflammatory illnesses, and drug-resistant infections [7]. These natural compounds include vitamins, phenolic compounds, and other bioactive compounds [8].

Phytochemicals are often referred to as antinutritional factors and they are usually not necessary for the body’s normal functioning but have an important role in therapeutic functions. It is well understood that antioxidants interfere with the development of free radicals. Free radicals can cause damage to cellular biomolecules such as nucleic acid, proteins, lipids, and carbohydrates, and can therefore adversely affect the immune system [9]. Natural plant extract can potentially provide new mechanisms of action for a new source of antimicrobial agents. The use of plant BLCs with established antimicrobial properties can be of great importance for therapeutic treatments [10]. According to the literature, there is a dearth of information about the potential use of *acacia* BLCs in the nutrition of rabbits. Therefore, the present study was an in vitro study evaluating the effect of *Acacia nilotica* bark BLCs (ANBBLCs) in regard to their antioxidant and antimicrobial activities, and further their effects on the productive performance, antioxidant status, and antimicrobial activities of growing rabbits under hot climatic conditions. 

## 2. Materials and Methods

### 2.1. Housing, Animals, and Design

The Department of Animal and Poultry Production, Faculty of Agriculture, South Valley University, Qena, Egypt, is where the animal experiment was carried out on rabbits. During the 56-day duration of this experiment, rabbits were housed in similar managerial, sanitary, and environmental circumstances. A total of 100 California male weanling rabbits aged 35 days were divided into four nutritional treatments, each of which contained ANBBLCs at concentrations of 0 (control group), 50, 100, and 150 mg/kg diet (*n* = 25 per treatment, each replication consisting of one animal). The average body weight [BW] of the animals was 613 ± 14 g. The South Valley University’s Animal and Poultry Production Department’s Committee of Ethics (SVU-AGRI-2-2022) issued its approval for the experiment, which ensured that the rabbits were treated throughout in accordance with the standards for the care of experimental animals. Rabbits were raised individually in metabolic cages made of galvanized metal wire nets (Co. D.R. Jow, Cairo, Egypt) that measured 50 cm in width by 60 cm in length by 40 cm in height, accompanied by manual feeders and drinkers. During the experiment, the rabbits were housed in an open system with an average temperature of 39.2 °C, a relative humidity of 55–60%, and a 16–8 h light–dark cycle. Inside each cage were stainless steel nipples that provided free access to fresh tap water. Table 1 shows the feed ingredients and approximate chemical analysis of the pelleted basal diet which was formulated at the Rabbit Farm, Department of Animal and Poultry Production, Faculty of Agriculture, South Valley University, Qena, Egypt.

### 2.2. Preparation of Acacia nilotica BLCs

The bark of the *Acacia nilotica* tree was recently harvested from a field in an agriculture farm in Qena, Egypt. Samples were physically and randomly collected, weighed immediately, and then dried in the sun for 48 h. Prior to use, dried materials were ground to pass through a 1 mm screen with a centrifugal mill (ZM1, Retsch, Haan, Germany) and kept at room temperature in airtight containers (plastic with tight-fitting lids). Weighing 5 g of plant powder and placing it into a 250 mL flask with 100 mL of 96% ethanol produced ANBBLCs. The extraction was carried out using an ultrasonic water bath (Bandelin electronic GmbH & Co. KG, Berlin, Germany) for two hours at 25 °C. The samples were filtered after extraction, and the solvent was then evaporated using a rotary evaporator (RVP100301, OMA Co., Everett Drive, West Windsor, NJ, USA) at 40 °C [8]. 

### 2.3. Growth Performance

At 35, 63, and 91 days of age, the rabbits’ body weights (BW) were recorded. By measuring feed residue on the same day, the feed intake for each pen between weighing was estimated. The weight of the feed consumed was divided by the daily BW gain per pen to determine the daily feed conversion ratio (FCR), which is defined as feed per gain. Diarrheal signs were noted every day, and deaths were noted when they happened.

### 2.4. Carcass Measurements

The rabbits for each treatment were weighed, and slaughtered at the end of the experimental period at the age of 90 days at the slaughterhouse of the Department of Animal and Poultry Production, Faculty of Agriculture of the South Valley University, Qena, Egypt. The animals were humanely slaughtered in accordance with the halal slaughter procedure. The procedure involved severing the carotid artery, jugular vein, trachea, and esophagus. The slaughtered rabbits were bled, and then the skin, genitals, urinary bladder, gastrointestinal tract, and the distal parts of the legs were removed. The carcasses were weighed with the following organs included: liver, heart, spleen, lungs, kidneys, as well as perirenal and scapular fat. The internal organ weights, including the liver, heart, head, kidneys, and spleen, were measured, and expressed as a percentage of the slaughter weight.

### 2.5. Determination of the Total Antioxidant Capacity

The phospho-molybdenum technique was used to assess the ANBBLCs’ total antioxidant capacity (TAC) [11]. Ascorbic acid was used as a reference antioxidant in the calculations, and the results were represented as mg of ascorbic acid equivalent per 100 g of extract (mg AAE/100 g). According to Cobanová et al. [12], the total antioxidant capacity of the blood and liver tissues was determined. To monitor and evaluate the accuracy of the analytical approach, we conducted a second analysis on the same sample after it had been independently prepared and examined for the parameters. The liver tissues were thawed at room temperature and homogenized with phosphate buffer saline (pH 6.8). The homogenized liver tissues were centrifuged at 13,000× *g* for 5 min at 4 °C in order to reduce the protein content that may affect the antioxidant capacity, and then were stored at −80 °C until analysis. Serum samples were obtained by collecting blood and allowed to clot for 30 min at 25 °C. Samples were then centrifuged at 3000 rpm for 15 min at 4 °C. The top yellow serum layer was separated into Eppendorf tubes and these were stored at −80 °C. The TAC in serum and liver tissues was determined using a commercial colorimetric assay (Bio-diagnostic, Cairo, Egypt) according to the manufacturer’s instructions. 

### 2.6. Determination of the Total Phenolic Content of Acacia Extracts

The Folin–Ciocalteu technique was used to determine the ANBBLCs’ total phenol content [13]. The results were calculated using the regression equation of the calibration curve and represented as mg of gallic acid equivalents per kilogram of the extract (mg/Kg). Using quercetin as a reference, the aluminum chloride test [14] was used to assess the total flavonoid content of the ANBBLCs. The results were calculated as mg quercetin equivalent/Kg of extract (mg/Kg).

### 2.7. Antibacterial Activity

The antibacterial activity of ANBBLCs in the rabbit cecum was determined following the procedure of Malabadi et al. [15]. The bacterial strains of *Bacillus subtilis*, *Escherichia coli*, *Staphylococcus* spp., and *Lactobacillus* spp. were used. The bacterial counts were measured at the Microbiology Unit in the Microanalytical Center, Cairo University. In order to count the colony forming units (CFU) of the *Bacillus subtilis*, *Escherichia coli*, *Staphylococcus* spp. and *Lactobacillus* spp. in the cecum, the cecum was promptly removed from the slaughtered rabbits. The cecum content samples were carefully mixed after being decanted into several sterile plastic containers. A stomacher bag was filled with ten grams of homogenized material, with ten-fold successive dilutions using physiological NaCl-Trypton, and the mixture was violently agitated for three minutes. MRS agar was used as the plate medium for *Bacillus subtilis*, and *Lactobacillus* spp. were counted by inoculating *E. coli* Petrifilms (3M Corporation, St. Paul, MN, USA) with a 10-fold serial dilution of rinses. To dilute the sample, sterile saline (0.85%) was used in accordance with the recommendations of the manufacturer. Typical *Escherichia coli* colonies were counted after a 24 h incubation period at 35 °C.

### 2.8. Statistical Analysis

The study was of a completely randomized design and the data generated were analyzed using SAS 9.2’s software, and the general linear models (GLM) technique was used to conduct the statistical analysis (SAS Institute, Cary, NC, USA, [16]). The only fixed factor in the model was the dose of supplementation. The animal served as the experimental unit for growth performance, carcass criteria, microbiological parameters, and antioxidant parameters. The data were tested for normal distribution (Anderson–Darling test for normality). To determine the linear and quadratic effects of increasing levels of ANBBLC supplementation, orthogonal polynomial contrasts were used, and Duncan’s multiple range test was used to compare means. Significance was declared at *p* < 0.05. *p*-values less than 0.001 were expressed as “<0.001” rather than the actual value.

## 3. Results

### 3.1. Total Antioxidant Capacity, Total Phenols, and Total Flavonoids in Acacia Bark BLCs

The total antioxidant capacity (TAC), total phenols (TP), and total flavonoids (TF) of ANBBLCs are given in Table 2. The highest values of TAC, TP, and TF using ethanol as an extraction solvent were detected in *Acacia* bark BLCs. The TP content could function as a key indicator of the antioxidant abilities of *Acacia* bark BLCs. The data of the present study indicate a highly positive correlation between the TAC and TP contents of all extracts (Figure 1). The correlation coefficient (R^2^) (0.86) indicates that the phenolic compounds in *Acacia* extracts contributed 86.2% of their antioxidant capacities. The phytochemical contents, especially polyphenols, are the main contributors to the total antioxidant capacity of ANBBLCs.

### 3.2. Productive Performance

Dietary ANBBLC levels linearly (*p* < 0.05) improved BW at 63 and 91 days of age and BW gain during 35–63, 63–91, and 35–91 days of age (Table 3) under hot climatic conditions. In comparison to the control group at ages 35–63, 63–91, and 35–91 days, the FCR values of growing rabbits were linearly improved with increasing concentrations of ANBBLCs (Table 3). The results show that feed intake was quadratically increased (*p* < 0.001) with the increase in dietary ANBBLC levels during 35–63 days of age. However, there were no differences in feed intake due to ANBBLCs during 63–91 and 35–91 days of age (Table 3). 

### 3.3. Carcass Criteria

The effects of ANBBLC treatments on the carcass parameters and various internal organs of growing rabbits are mentioned in Table 4. In contrast to the control group, the perirenal and scapular fat to BW ratio linearly decreased (*p* < 0.001) after ANBBLC supplementation, and the dressing percentage increased (*p* = 0.010) under hot climatic conditions. The liver, heart, spleen, kidney, and head of growing rabbits showed no alterations when ANBBLCs were added into their diets. 

### 3.4. Blood and Liver Tissue Total Antioxidant Capacity (TAC)

The dietary supplementation with ANBBLCs showed a similar trend in TAC in blood serum and liver tissues compared to the control group (Figure 2A,B). With increasing concentrations of ANBBLCs, the TAC of the liver tissues of growing rabbits (Figure 2A) was linearly (*p* < 0.001) enhanced. Likewise, supplementations of ANBBLCs linearly (*p* < 0.001) improved serum TAC compared to the control (Figure 2B). 

### 3.5. Cecum Microbial Counts

*Lactobacillus* spp. counts were linearly (*p* < 0.001) increased with the increase in dietary ANBBLC concentrations compared to the control (Figure 3A). However, *Staphylococcus* spp. counts were linearly (*p* < 0.001) decreased in comparison to the control group when ANBBLCs were added at different levels in the diets of rabbits (Figure 3B). On the other hand, *Escherichia coli and Pseudomonas* spp. and *Bacillus subtilis* and *Staphylococcus* spp. were not detected in any cecum groups under hot climatic conditions.

## 4. Discussion

There is a dearth of information in the literature regarding the potential use of ANBBLCs in rabbit feeding and the effects on the rabbits’ antioxidant and antibacterial properties. Total phenolic content may serve as a key indicator of ANBBLCs’ antioxidant abilities [17]. The results of this investigation showed a strong relationship between ANBBLC extracts’ total antioxidant activity and total phenolic contents. 

The phenolic components in *Acacia* extracts contributed 86.2% to their antioxidant capabilities, according to the correlation value (R^2^) (0.86). The primary reason for ANBBLCs’ overall TAC is due to their phytochemical components, particularly their polyphenol concentrations [18]. 

In this study, the growth performances in terms of BW, BW gain, and FCR were improved with the increase in dietary ANBBLC concentrations under hot climatic conditions, partly attributed to a tendency of increasing feed intake. The ability of ANBBLCs to improve growth performance and stimulate appetite and feed intake is mainly due to its phytochemical components, particularly its polyphenol concentrations [17,18]. The major constituents of ANBBLCs were 13-docosenoic acid (34.06%), lupeol (20.15%), 9,12-octade-cadienoic acid (9.92%), and 6-octadecanoic acid (8.43%), according to the previous study of Ali et al. [19]. We compared the results with previous studies on herb or plant BLCs because, as far as we know, there is no publication that discusses ANBBLCs and their effects on growing rabbits. The favorable benefits of herbs, extracts, or active ingredients in animal nutrition have been documented to promote appetite and feed intake, which in turn improves growth performance [20]. This can be explained by the beneficial effects of plant BLCs, which include the intestinal colonization of beneficial bacteria, ideal intestinal flora composition, efficient nutrient absorption, and consequently good health and productive performance [21]. The fact that plant BLCs are effective antioxidants and scavenge oxygen free radicals, which are damaging byproducts of numerous metabolic processes, can be used to explain why rabbits fed ANBBLCs performed better in terms of growth [22].

In addition, in the current investigation, the total antioxidant capacity of blood and liver tissue was linearly increased with increasing concentrations of ANBBLCs under hot climatic conditions. *Acacia* BLC treatments significantly improved physiological parameters and the antioxidant capacity of growing rabbits, and this improvement was dose dependent. Similar results were reported by Saleem et al. [23], who stated that *Acacia nilotica* extracts were more potent antioxidants than alpha-tocopherol. According to Dafallah and Al-Mustafa [24], the aqueous *Acacia nilotica* extracts contain phytoconstituents such as flavonoids, polysaccharides, and organic acids that may be primarily responsible for their pharmacological actions. According to Fran et al. [22], flavonoids (isoflavones) capture radicals before they can damage a cell, quenching active singled oxygen, and decomposing hydrogen peroxide without creating radicals. The *Acacia* seed extract-treated rabbits had higher levels of antioxidants such as catalase, SOD, and GSH, which are used to scavenge and destroy free radicals, while also having lower levels of free radicals [25]. 

When *Acacia* bark BLC levels were added to the diets of rabbits in this study, *Lactobacillus* spp. was enhanced, but *Staphylococcus* spp. was lowered in comparison to the control group under hot climatic conditions. Consequently, it is clear from the research that phytogenic BLCs have strong antibacterial properties and can be added to animal feed to boost development or ward against sickness. ANBBLCs should be considered to be a potential candidate as an alternative to antibiotic growth promoters in rabbit diets based on our results, but further systematic studies are warranted. The antibacterial activity of plant extracts is usually related to their phytochemical compounds such as phenols, flavonoids, alkaloids, tannins, and glycosides [26]. This is in line with the findings of other authors, including Ali et al. [27] and Chattopadhyay et al. [28]. Although their antimicrobial effects in animal nutrition and health have been extensively studied, there is still much to learn about the mechanisms by which plant BLCs exert their antimicrobial effects on various microbial populations, on target sites, in a feed matrix, and in the presence of other phytochemicals with competing effects. The effects of plant BLCs to prevent diseases are well established. Plant BLCs may preferentially target bacteria, which could be a beneficial enhancement to the eubiosis [29]. In the current study, the increase in growth performance is the outcome of improved nutrient use and absorption mainly due to the beneficial effects on the digestive system by a variety of plant BLCs, including those with antibacterial characteristics. The presence of phytochemicals such as flavonoids, terpenoids, and phenols, which have been reported to have antioxidant characteristics [30], may be responsible for these biological activities of *Acacia nilotica* including growth performance and carcass criteria under hot climatic conditions in this study. Furthermore, in various in vitro experiments, kaempferol, a polyphenolic component identified in *Acacia nilotica*, showed a radical scavenging ability [31]. By inhibiting the cyclooxygenase enzymes involved in the inflammatory process, niloticane, a compound isolated from the bark of *Acacia nilotica*, demonstrated anti-inflammatory properties [32]. *Acacia nilotica* has demonstrated significant antioxidant activity because of its high concentration of polyphenolic elements, such as catechins, which are known to have antioxidant and anti-inflammatory effects [33], and therefore improve the animal’s ability to alleviate the effects of hot climatic conditions. Furthermore, ethyl gallate extracted from the ethanolic extract of *Acacia nilotica* BLCs has been demonstrated to have antioxidant properties by Kalaivani et al. [34]. According to Kannan et al. [35], *A. nilotica* treatment caused liver enzyme levels to return to normal ranges following acetaminophen-induced liver injury. This indicates that *A. nilotica* has hepatoprotective properties. Additionally, *A. nilotica* extracts were shown to be protective against hepatic damage brought on by 2-butoxyethanol in male mice [36]. Moreover, the activities of AST, ALT, and ALP were normalized, indicating the plant’s hepatoprotective capacity [37]. The administration of *Acacia nilotica* BLCs (400 mg/kg BW/day for 30 consecutive days) was able to protect the liver organs from the adverse effects brought on by atrazine exposure [38].

In the current trial, adding ANBBLCs to rabbit diets significantly increased dressing percentages and decreased perirenal and scapular fat under hot climatic conditions. Overall, the percentages of organs, yields, dissectible fat, and carcass weight were comparable to those observed in earlier environments [39,40]. It seems that the lack of quantitative information about *Acacia* bark BLCs also hinders advancement in terms of carcass characteristics, but studies to determine this should be conducted in future.

## 5. Conclusions

The results of the present investigation, which indicate a positive influence of feed supplementation with *Acacia nilotica* bark BLCs on growth performance, carcass criteria, and antioxidant capacities, serve as a basis for the presumed beneficial use of *Acacia nilotica* bark BLCs as an effective feed additive. To increase the use of plant extracts in food and medicine, additional studies should be planned to examine the separation and characterization of bioactive components from *Acacia*, and higher doses should be tested for any additional benefits.

## Figures and Tables

**Figure 1 animals-13-01933-f001:**
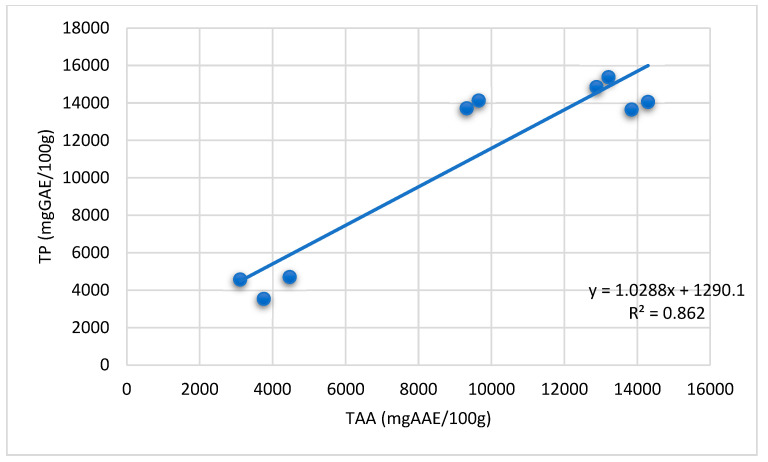
Relationship between the total antioxidant activity and phenolic content of the *Acacia* BLCs.

**Figure 2 animals-13-01933-f002:**
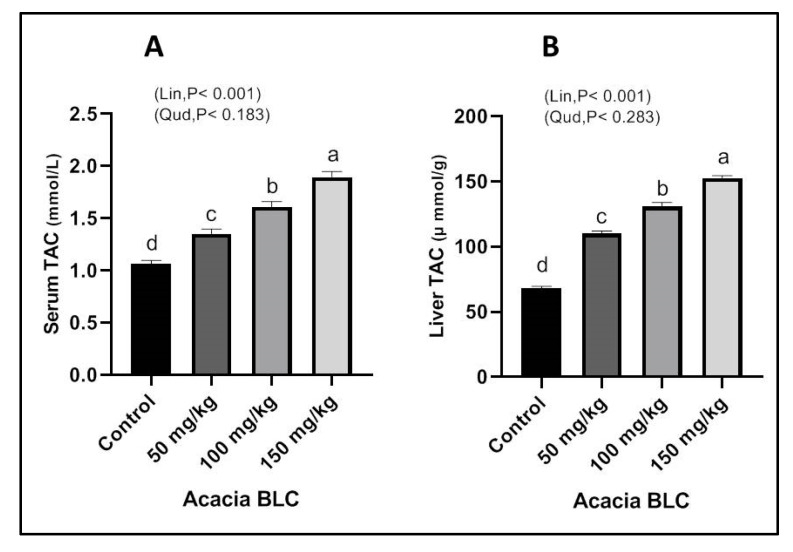
Impact of *Acacia* bark BLC supplementation on total antioxidant capacity (TAC) in blood (**A**) and liver tissues (**B**) of rabbits at 91 days of age. Bars with different letters (a, b, c, d) are significantly different (*p* < 0.05).

**Figure 3 animals-13-01933-f003:**
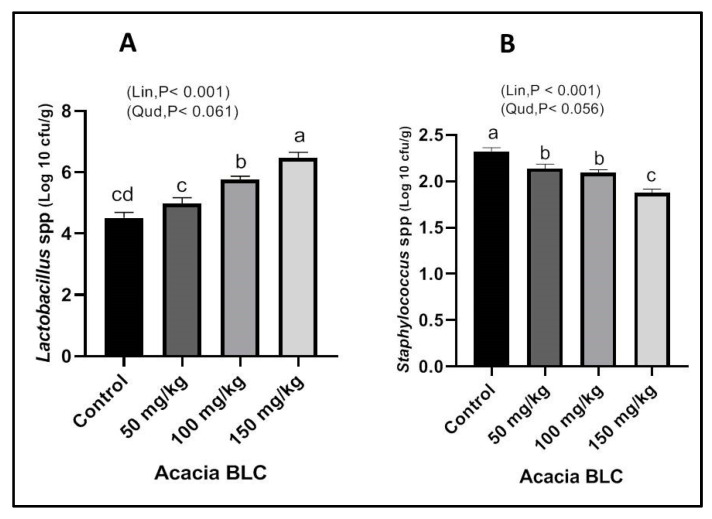
Impact of *Acacia* bark BLC supplementation on cecal bacterial counts (*Lactobacillus* spp. (**A**) and *Staphylococcus* spp. (**B**)) of rabbits at 91 days of age. Bars with different letters (a, b, c, d) are significantly different (*p* < 0.05).

**Table 1 animals-13-01933-t001:** Ingredient composition (as-fed basis) of the basal diet.

Ingredients	g/kg	Chemical Composition Analyzed	(g/kg, as Fed)
Corn	310	Dry matter	936
Wheat bran	200	Gross Energy (MJ/kg DM)	17.3
Soybean meal (440 g/kg CP)	190	Crude protein	178
Wheat straw	120	Ether extract (fat)	39.7
Lucerne hay	60	aNDFom	392
Rice bran	40	ADFom	232
Linseed straw	28	ADL	69.5
Sunflower meal	25	Ash	92.82
Limestone	20	Calcium	15.80
Sodium chloride	3	Phosphorus	7.73
Vitamin-mineral premix ^1^	3	Zinc	45
Dl-Methionine	1		

^1^ Per kg of diet: vitamin A 10,000 IU, vitamin D3 900 IU, vitamin E 50.0 mg, vitamin K 2.0 mg, vitamin B1 2.0 mg, folic acid 5.0 mg, pantothenic acid 20.0 mg, vitamin B6 2.0 mg, choline 1200 mg, vitamin B12 0.01 mg, niacin 50 mg, biotin 0.2 mg, Cu 0.1 mg, Fe 75.0 mg, Mn 8.5 mg, ZnO 20 mg. aNDFom: neutral detergent fiber, ADFom: acid detergent fiber, ADL: acid detergent lignin.

**Table 2 animals-13-01933-t002:** Antioxidant capacity, total phenols, and total flavonoids activity of *Acacia* bark BLCs.

Properties	Ethanol Extract
Total antioxidant capacity (mg AAE/100 g)	1321 ± 56
Total phenols (mg GAE/100 g)	15,376 ± 85
Total flavonoids (mg QE/Kg)	513 ± 6

AAE: ascorbic acid equivalents, GAE: gallic acid equivalents, QE: quercetin equivalents.

**Table 3 animals-13-01933-t003:** Effects of *Acacia* BLCs on growth performance of growing rabbits.

Items	*Acacia* BLC Levels (mg/kg)				SEM	*p*-Value	
	0	50	100	150		Lin	Quad
Body weight							
35 days	675	664	643	673	19.08	0.768	0.296
63 days	1470 ^b^	1594 ^a^	1600 ^a^	1650 ^a^	26.63	0.001	0.179
91 days	2426 ^c^	2620 ^b^	2712 ^b^	2852 ^a^	53.04	<0.001	0.618
Body weight gain							
35–63 days	28.39 ^b^	33.23 ^a^	34.18 ^a^	34.87 ^a^	0.988	0.001	0.052
64–91 days	34.14 ^c^	36.61 ^b^	39.71 ^a^	42.93 ^a^	1.856	0.003	0.844
35–91 days	62.54 ^b^	69.84 ^a^	73.89 ^a^	77.8 ^a^	1.923	<0.001	0.389
Feed intake							
35–63 days	86.36 ^b^	91.07 ^a^	90.86 ^a^	89.32 ^a^	1.021	0.075	0.007
64–91 days	98.39	97.28	99.98	99.21	1.333	0.398	0.899
35–91 days	184.7	188.4	190.8	188.5	1.663	0.081	0.095
Feed conversion ratio							
35–63 days	3.072 ^a^	2.743 ^b^	2.659 ^b^	2.572 ^b^	0.094	0.002	0.221
64–91 days	2.949 ^a^	2.695 ^b^	2.520 ^b^	2.328 ^c^	0.161	0.013	0.849
35–91 days	2.972 ^a^	2.708 ^b^	2.584 ^c^	2.426 ^c^	0.076	<0.001	0.499

SEM, standard error of the means (*n* = 25, individual rabbits per diet). Lin, linear responses to dietary inclusion levels of ANBBLCs. Quad, quadratic responses to dietary inclusion levels of ANBBLCs. ^a,b,c^ means with different superscript are significantly different (*p* < 0.05).

**Table 4 animals-13-01933-t004:** Effects of *Acacia* BLCs on carcass criteria of growing rabbits.

Items	*Acacia* BLC Levels (mg/kg)				SEM	*p*-Value	
	0	50	100	150		Lin	Quad
LBW, g	2660	2720	2732	2652	43.93	0.945	0.132
Dress,%	55.08 ^b^	58.45 ^a^	57.63 ^a^	59.88 ^a^	1.04	0.010	0.599
Liver,%	2.58	2.83	2.82	2.59	0.20	0.971	0.248
Heart,%	0.367	0.323	0.272	0.297	0.02	0.741	0.541
Head,%	4.240	5.274	5.152	5.501	0.58	0.175	0.562
Spleen,%	0.033	0.047	0.046	0.047	0.007	0.190	0.379
Kidney,%	0.636	0.653	0.695	0.578	0.039	0.468	0.111
Abdominal fat,%	1.636 ^a^	1.158 ^b^	1.131 ^b^	1.069 ^b^	0.066	<0.001	0.007

SEM, standard error of the means (*n* = 25, individual rabbits per diet). Lin, linear responses to dietary inclusion levels of ANBBLCs. Quad, quadratic responses to dietary inclusion levels of ANBBLCs. ^a,b^ means with different superscript are significantly different (*p* < 0.05).

## Data Availability

The datasets generated and/or analyzed during the current study are available from the corresponding author on reasonable request.

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
