# Peer review of "Effects of Dietary Bioactive Lipid Compounds of Acacia nilotica Bark on Productive Performance, Antioxidant Status, and Antimicrobial Activities of Growing Rabbits under Hot Climatic Conditions"

_animals, 2023, doi:10.3390/ani13121933_

Round 1
Reviewer 1 Report
Summary of Comments on animals-2430442-peer-review-v1.
Title: Effects of dietary bioactive lipid compounds of Acacia nilotica bark on productive performance, antioxidant status, and anti- microbial activities of growing rabbits under hot climatic conditions.
Summary:
This is an interesting study using phytochemicals as alternative to at conventional feed additives or antimicrobials to improve gut health, production parameters and meat yield and quality.
Line 67-68: citation
Line 94: write hour in full.
Line 108: How long were the plant sun-dried?
Line 183 and 184: can change indicated to "indicates."
Line 200: use lower case q for quadratically.
Table 3, 4: can you provide the superscripts for the data with significant p-values?
You have multiple places where you use in this current study, you can change some of them to "In this study----"
What other farm animals has this product been tested on? Poultry?
This manuscript is well written.
Author Response
Dear Respective Reviewer, thank you very much for your positive opinion of our manuscript and inputs which has improved the final form of our manuscript. Please find our reply to your comments below and a list of changes that we made according to your suggestions. We have addressed all the suggestions in the revised manuscript. The authors’ responses are in the attached files.

Reviewer 2 Report
Title: Effects of dietary bioactive lipid compounds of Acacia nilotica barkon productive performance, antioxidant status, and antimicrobial activities of growing rabbits under hot climatic conditionsAuthors: Ahmed A.A. Abdel-Wareth *, Hazem G.M. El-Sayed, Hamdy A. Hassan,Ghadir El-Chaghaby, Abdel-Wahab A. Abdel-Warith, Elsayed M. Younis, Shimaa A.Amer, Sayed Rashad, Jayant Lohakare
Thank you for inviting me to review the paper by Ahmed A.A. Abdel-Wareth et al. Digestive disorders are the main factor responsible for reduced performance and health in growing rabbits. Rabbit breeding is an important enterprise in many countries, because these animals provide high quality meat for human consumption. To achieve better production by rabbit husbandry, one possible way is to use natural compounds, such as probiotics, prebiotics, synbiotics, bacteriocins, organic and fatty acids, and plant extracts, which can improve the health and well-being of animals. There are many reviews discussing the beneficial effects of such additives, mostly supplementation of probiotics to rabbits, with emphasis on reduction of pathogens and parasites in the gastrointestinal tract, stimulation of immunity, better growth performance, metabolism and nutrient utilization, and improved meat quality. The use of natural antimicrobials, including probiotics, bacteriocins, and herbal extracts, can modulate and balance the intestinal microbiota substituting the role of beneficial lactic acid bacteria, and enhance the immune response. The mechanism of action of some antimicrobials, such as probiotics is well known: Reduction of toxin production, stimulation of enzyme production by the host, competition for adhesion to epithelial cells, stimulation of the immune system of the host (mainly due to stimulation of gut‐associated lymphoid tissue and mucosa‐associated lymphoid tissue, decreasing intestinal pH, and production of vitamins (B, K) and antimicrobial substances (lactic and acetic acids, hydrogen peroxide, carbon dioxide, diacetyl, and bacteriocins. These are the reasons why this topic is highly relevant.
The aim of the submitted study: "Effects of dietary bioactive lipid compounds of Acacia nilotica bark on productive performance, antioxidant status, and antimicrobial activities of growing rabbits under hot climatic conditions" was to evaluate the effectiveness of bioactive compounds found in the bark of Acacia nilotica (ANBBLC) used as new feed additives on the growth performance, criteria of slaughter rabbit bodies, antioxidants and antimicrobial activity of growing rabbit males.
This is a pilot study as the potential use of ANBBLC as a supplement to rabbit feed has not yet been published. As the authors investigated only some aspects of the positive effect of ANBBLC on the health of rabbits, below is a list of all the items that need to be addressed in this study.
In the Part 2. Materials and Methods the following adjustments should be made:
· · Specify type and manufacturer of metabolic cages
· · Specify type and producer of the basal diet
· · Dried samples were ground into particle size of less than 1 mm and stored in air-tight containers at room temperature until use. Specify instrument (type, producer) used for grinding the samples
· · Airtight containers – provide more detailed description
· · In Table 1, Ingredients composition (as-fed basis) of the basal diet - add information about amount of fat in the basal diet
· · Ultrasonic water bath – type, manufacturer?
· · After extraction the samples were filtered, and the solvent was evaporated using a rotary evaporator at 40°C – supply type and manufacturer of rotary evaporator
· · It is reported that animals were slaughtered humanely - provide detailed description of the slaughtering process
· · The slaughtered rabbits were bled, - from where was the blood drawn, amount, vessel used for collection?
· · The bacteria strains of the Bacillus subtilis, Escherichia Coli, Staphylococcus spp. and Lactobacillus spp. were measured, should be correct to: Bacillus subtilis, Escherichia coli, Staphylococcus spp. and Lactobacillus spp. were measured.
· · It is necessary to present details about media and conditions that were used for determination of total counts of investigated bacteria.
In the part 3. Results the following adjustments are needed:
· - correct Total phenols to Total Phenols
· - correct AAE: ascorbic acid equivalents to AAE: Ascorbic acid equivalents
In the part 5. Conclusions the following is recommended:
· Conclusions of authors regarding the use of Acacia nilotica bark BLC as an effective feed supplement for improvement of growth, criteria of slaughter carcasses and antioxidant capacity as well as antimicrobial activities in growing rabbits based only on the pilot study are overestimated. In particular, conclusions derived from the results of antimicrobial activities, obtained by monitoring of proportions of selective bacterial representatives by cultivation methods are not justified. Arriving to such conclusion would require to analyse the total caecal microbiome by means of already commonly available more accurate analytical methods, such as next-generation sequencing analysis. Therefore, in conclusion, I recommend to only summarize briefly the obtained results and state that these results, indicating a positive influence of feed supplementation with Acacia nilotica bark BLC (on the investigated productive parameters) serve as a basis for presumed beneficial use of Acacia nilotica bark BLC as an effective feed additive.
· · If additional experiments are to be performed, then the morphometry and functionality of the intestinal mucosa and the composition of the caecal microbiota should be monitored for a more objective assessment of animal growth performance and food digestibility, as it is common in studies with a similar focus.
Author Response
Dear Respective Reviewer, We sincerely appreciate your positive opinion, comments, and inputs which helped to improve our manuscript's final form and your favourable review of our work. Please find our reply to your comments below and a list of changes that we made according to your suggestions. We have addressed all the suggestions in the revised manuscript. The author’s responses are in the attached files.

Reviewer 3 Report
The manuscript “Effects of dietary bioactive lipid compounds of Acacia nilotica bark on productive performance, antioxidant status, and anti-microbial activities of growing rabbits under hot climatic conditions” by Abdel-Wareth et al., aims to evaluate the efficacy of dietary Acacia nilotica bark bioactive lipid compounds (ANBBLC) as novel feed additives on growth performance, carcass criteria, antioxidants and antimicrobial activities of growing male rabbits using 100 35-day-old weanling male rabbits.
It is true that rabbits are highly susceptible to high temperatures which negatively affect their metabolism, performance, carcass metrics, and meat quality. Supplementation of plant bioactive lipid compounds in feed is an efficient way to improve the health, growth, and productivity of rabbits. Potential of natural compounds derived from plants is immense. Acacia nilotica tree is well known in traditional medicine and many bioactive secondary compounds have been identified in this plant. Not much work has been done on the potential of the tree in the nutrition of rabbits. Therefore, the authors planned the current work to explore the potential of the plant as feed additive for rabbits. The gaps in knowledge and objectives of the study are very clear and the authors addressed an important issue in meat rabbit farming. In that context, the current work has merit to add new knowledge to the existing knowledge pool in rabbit nutrition. The manuscript is well written and easy to understand. For further improvement of the manuscript, the authors are requested to clarify the following points.
1. The authors included ANBBLC at concentrations of 0 (control group), 50, 100, and 150 mg/kg diet. I want to know on what basis the dietary treatments were designed? Based on any preliminary experiment or any previous literature? As per results, 150 mg/kg diet supplemented group showed best results. Why the authors didn’t increase the dose little further? The authors are requested to clarify.
2. It will be interesting to what are the bioactive lipid compounds present in Acacia nilotica bark extract. That may provide insight on the beneficial effects of the extract. The authors evaluated total phenols (TP), and total flavonoids (TF) in the extract but it would have been better if the authors could estimate bioactive lipids and their derivatives.
3. The authors evaluated the total antioxidant capacity (TAC) in serum and liver following supplementation of ANBBLC in diet. The authors are requested to estimate major antioxidant enzymes like superoxide dismutase (SOD), catalase (CAT), and glutathione peroxidase (GPX) as well as level of lipid peroxidation which may give clear picture about the antioxidant property of ANBBLC.
4. The authors justified the growth promoting, antioxidant and antimicrobial properties of the extract are due to its phytochemical components present in the extract. I would like to know what are the phytochemical components present in the extract and how they are contributing to the above-mentioned properties. That may give more scientific perspective to the manuscript.
The English Language is good except few minor typos.
Author Response
Dear Respective Reviewer, thank you very much for your positive opinion of our manuscript and inputs which improved the final form of our manuscript. Please find our reply to your comments below and a list of changes that we made according to your suggestions. We have addressed all the suggestions in the revised manuscript as track changes. The author’s responses are in the attached files.

Round 2
Reviewer 3 Report
The authors addressed all the concerns and the manuscript is improved significantly. I am satisfied with the response. The authors mentioned that they could not measure the concentration of the antioxidant enzymes and would take up in future study. I think that is acceptable. Overall, the revised manuscript looks good and it contains novel information. I feel that the manuscript has the potential to be published in Animals.
The English language looks fine.